# TRP Channels in Skin Cancer: Focus on Malignant Melanoma

**DOI:** 10.3390/ijms26167829

**Published:** 2025-08-13

**Authors:** Damian Twardak, Vita Havryliuk, Maciej Gagat

**Affiliations:** 1Department of Histology and Embryology, Faculty of Medicine, Nicolaus Copernicus University in Toruń, Collegium Medicum in Bydgoszcz, 85-092 Bydgoszcz, Poland; damian.twardak@cm.umk.pl (D.T.); vita.havryliuk@cm.umk.pl (V.H.); 2Department of Morphological and Physiological Sciences, Faculty of Medicine, Collegium Medicum, Mazovian Academy in Płock, 09-402 Pock, Poland

**Keywords:** transient receptor potential (TRP), melanoma, TRPM1, TRPM7, TRPM8, TRPV1, TRPV4

## Abstract

Cutaneous malignant melanoma remains one of the most aggressive forms of skin cancer, characterized by high metastatic potential and resistance to standard therapies. Emerging evidence suggests that transient receptor potential (TRP) channels, non-selective cation channels involved in calcium homeostasis, and cellular stress responses play a pivotal role in melanoma development and progression. This review highlights the physiological expression of key TRP subfamilies (TRPM1, TRPM7, TRPM8, TRPV1, TRPV4, and TRPM2) in melanocytes and discusses their dysregulation in melanoma cells. TRPM1 is implicated as a tumor suppressor, whereas TRPM7, TRPV1, and TRPV4 often function as both melanoma suppressor or oncogenic drivers, modulating proliferation, apoptosis, and metastasis. TRPM2, which is responsive to oxidative stress, supports melanoma cell survival under metabolic stress. The potential of TRP channels as diagnostic biomarkers and therapeutic targets is evaluated, with attention paid to current pharmacological approaches and research challenges. The complexity and context-dependency of TRP function in melanoma underscore the need for isoform-specific modulation and personalized therapeutic strategies.

## 1. Introduction

Melanoma is a malignant tumor that originates from melanocytes, which are derived from neural crest. Although melanomas most commonly arise in the skin, they can also develop in other locations where neural crest cells migrate, such as the gastrointestinal tract and brain [1]. The American Cancer Society estimates that nearly 105,000 new cases of invasive melanoma of the skin and over 107,000 new cases of melanoma in situ will be diagnosed in 2025. As expected, early-stage melanoma confined to the primary site is associated with a highly favorable outcome, with over 99% of patients surviving five years. However, once the disease spreads to regional lymph nodes or distant organs, the five-year survival drops markedly to around 75% and 35%, respectively. Cutaneous malignant melanoma is one of the most aggressive and lethal forms of skin cancer, characterized by its high metastatic potential, resistance to conventional therapies, and significant heterogeneity at the molecular level. While advances in early detection, immunotherapy, and targeted treatment have improved outcomes in recent years, metastatic melanoma continues to present major clinical challenges, with limited long-term survival in many patients. Therefore, a deeper understanding of the molecular and cellular mechanisms driving melanoma progression is essential to identify novel diagnostic and therapeutic targets [2,3,4].

Among the molecular regulators implicated in tumor biology, growing attention has been directed toward the family of transient receptor potential (TRP) channels. TRP channels are a diverse group of non-selective cation channels that mediate the influx of calcium (Ca^2+^), sodium (Na^+^), and other ions across cellular membranes in response to a variety of chemical and physical stimuli [5]. Functionally, they act as cellular sensors of temperature, oxidative stress, mechanical forces, and osmotic changes, and they are involved in key physiological processes such as cell proliferation, differentiation, migration, apoptosis, and inflammation, all of which are frequently dysregulated in cancer [6,7]. In the context of melanoma, several TRP channel subtypes, including TRPM1, TRPM7, TRPV1, and TRPV4, have been shown to be differentially expressed and functionally active in tumor cells [8]. These channels have been linked to enhanced invasiveness, metastatic capacity, and modulation of the tumor microenvironment. Moreover, TRP channels are emerging as potential biomarkers for melanoma progression and candidates for targeted therapies due to their surface localization and regulatory influence on signaling pathways central to oncogenesis [9]. This review aims to summarize current knowledge on the expression and function of TRP channels, with a particular emphasis on their role in malignant melanoma of the skin. The present study focuses on their involvement in tumor progression. The potential of TRP proteins as diagnostic markers and therapeutic targets is also discussed, along with current limitations and future research directions.

## 2. Characteristics of TRP Channels

Transient receptor potential (TRP) channels, particularly those permeable to calcium ions, have emerged as critical modulators of intracellular calcium dynamics in various types of tumor cells. The expression profiles of these genes differ significantly across different types of malignancy, reflecting their context-dependent roles in cancer biology [8,10,11]. The TRP channel superfamily comprises several subgroups, including TRPA (Ankyrin), TRPV (Vanilloid), TRPC (Canonical), TRPP (Polycystin), TRPM (Melastatin), and TRPML (Mucolipin) [12,13,14]. Each subfamily exhibits unique activation mechanisms and physiological functions. For instance, TRPA channels are primarily sensitive to mechanical stress, while TRPV channels respond to noxious stimuli, elevated temperatures, and inflammatory signals [15]. TRPC channels contribute to endothelial permeability, neural development, and the regulation of vascular smooth muscle and endothelial cell proliferation. TRPP channels, on the other hand, are involved in mechanosensation as well as the regulation of follicular maturation and differentiation [5,16]. The TRPM subfamily has been implicated in processes such as insulin secretion, allergic responses, and the migration of mast cells and dendritic cells. Lastly, TRPML channels play essential roles in vesicle trafficking and the maintenance of lysosomal pH homeostasis [17]. Given this remarkable functional diversity, TRP channels exert a profound influence on both physiological and pathological processes. In the tumor microenvironment—characterized by hypoxia, acidosis, and increased mechanical pressure—mechanosensitive TRP channels are activated in response to environmental stressors, including low pH and oxygen deprivation, thereby influencing tumor cell behavior. Notably, oxygen depletion can trigger TRP-mediated calcium influx, further promoting oncogenic signaling cascades. Changes in TRP channel expression often accompany oncogenic transformation [18,19,20]. Both the upregulation and downregulation of specific TRP isoforms can facilitate tumor progression by modulating calcium-dependent pathways that control proliferation, migration, survival, and resistance to apoptosis. Altered TRP channel activity may enhance the formation of functional homo- or heteromeric channel complexes, further amplifying these effects. Consequently, TRP channels are increasingly regarded as potential molecular markers of tumor stage and aggressiveness. In this context, TRP channels have been implicated in virtually all steps of the metastatic cascade, including tumor cell migration, angiogenesis, immune evasion, and colonization of distant organs. This review highlights their multifaceted contributions to melanoma progression and metastasis and discusses the potential of TRP channels as diagnostic markers, therapeutic targets, and prognostic indicators. Finally, we explore the current landscape of TRP-targeting pharmacological agents and their potential clinical applications in oncology.

## 3. Physiological Role and Expression of TRP Channels in Melanocytes

TRP channels are widely expressed in various skin cell types, including keratinocytes, melanocytes, fibroblasts, immune cells, and vascular endothelial cells, where they contribute to skin homeostasis and responses to environmental stimuli such as UV radiation, temperature changes, and mechanical stress [7,21]. In melanocytic cells specifically, TRP channels regulate key physiological processes such as melanogenesis, proliferation, differentiation, and calcium signaling, all of which are essential for maintaining pigmentary balance and skin integrity [22,23]. Several TRP subfamily members, especially those from the melastatin (TRPM) family, are involved in the regulation of melanocyte physiology, and some of them are also involved in the development of malignant melanoma [23,24].

TRPM1 plays a crucial role in melanocyte physiology by regulating both calcium signaling and melanin production. This ion channel has been demonstrated to mediate calcium influx, a process that is imperative for the transfer of melanosomes to keratinocytes, thereby augmenting skin protection against ultraviolet (UV) radiation [25]. The activation of TRPM1 has been shown to elevate intracellular calcium levels, a process that is critical for the synthesis and distribution of melanin [26]. Furthermore, TRPM1 expression correlates with melanin content in melanocytes, thereby underscoring its importance in pigmentation regulation [27]. Mutations or dysregulation of TRPM1 have been associated with pigmentation disorders, thus highlighting its role in maintaining normal pigmentary function [28]. Collectively, these findings illustrate the multifaceted role of TRPM1 in maintaining melanocyte homeostasis and skin integrity.

TRPM7 is a distinctive member of the TRPM subfamily, exhibiting dual functionality as both a cation channel and a protein serine/threonine kinase [29]. It has been demonstrated to facilitate the conduction of both monovalent and divalent cations, exhibiting a pronounced selectivity for divalent ions. TRPM7 has been shown to be permeable not only to calcium (Ca^2+^) and magnesium (Mg^2+^) but also to other divalent cations, including zinc (Zn^2+^), [30] nickel (Ni^2+^), and barium (Ba^2+^) [30]. It is ubiquitously expressed and plays a central role in the regulation of intracellular magnesium homeostasis [31].

TRPM8 is a cold receptor that responds to temperature stimuli, primarily located in peripheral sensory nerves [32]. In addition to low temperatures, the receptor is also activated in response to the application of menthol, icilin, or other cooling agents. Research undertaken on human melanocytes has demonstrated that the activation of TRPM8 results in an influx of calcium ions. These channels have been demonstrated to reduce pigment production activity in melanocyte cells both in vivo and in vitro. This phenomenon occurs through mechanisms that result in a decrease in the expression of other proteins, such as tyrosinase and tyrosinase-related protein (TRP)-1 [33]. The results obtained suggest that TRPM8 may act as a negative regulatory factor in relation to pigmentation in melanocytes.

The transient receptor potential vanilloid 1 (TRPV1) is a non-selective calcium-permeable cation channel. Various thermal, mechanical, and chemical stimuli can activate it [32]. The expression occurs in various cell types, including those of peripheral sensory neurons and skin cells. It has been demonstrated to play critical roles in a number of conditions, such as pain, itch and inflammatory disorders [34,35]. In human melanocytes, TRPV1 is involved in UV-induced intracellular calcium signaling, which modulates melanin synthesis. The activation of this process has been demonstrated to contribute to the process of melanogenesis. In addition, studies have shown that the inhibition of TRPV1 can result in a reduction in melanin production [36]. It has been reported that the activation of TRPV1 by chemical ligands, such as capsaicin, has been associated with an elevation in intracellular calcium levels without a significant effect on melanogenesis. This suggests the hypothesis that the TRPV1-mediated calcium influx may require specific contextual signals to induce melanin production [37].

Another ion channel belonging to the family of non-selective transmembrane TRP cation channels is TRPV4. It is expressed in most tissues in a variety of locations throughout the body, with particularly high levels found in skin cells, such as keratinocytes and melanocytes [38]. The channel exhibits a response to temperature as well as a range of mechanical and chemical stimuli, including those derived from UVB (type B ultraviolet rays) [39]. The activation of this channel has been demonstrated to facilitate the influx of calcium ions into melanocytes. An increase in the intracellular concentration of Ca^2+^ has been demonstrated to affect processes such as cell survival, response to oxidative stress, and the regulation of the activity of enzymes involved in melanogenesis. This indicates the important role of TRPV4 in maintaining cellular homeostasis [40].

It has been evidenced that several members of the TRP channel family are differentially expressed not only in normal melanocytes but also in melanoma cells, where they may acquire novel or deregulated functions [26]. While TRP channels regulate processes such as calcium homeostasis, melanin production, and oxidative stress responses under physiological conditions, increasing evidence suggests that their dysregulation may contribute to malignant transformation, tumor progression, and metastasis in melanoma. Altered expression patterns and functional changes in TRPM1, TRPM7, TRPM8, TRPV1, and TRPV4 have notably been found to modulate cell survival, apoptosis, proliferation, and migration in melanoma models [6,13,41]. Therefore, in the next section, we focus on the role of TRP channels in melanoma, exploring how their altered activity contributes to tumorigenesis.

## 4. TRP Channels and Melanoma Development and Progression

The development and progression of cutaneous malignant melanoma involve complex molecular pathways, including dysregulated ion channel activity [42]. Among these, TRP channels have emerged as important modulators of tumorigenesis due to their roles in calcium homeostasis, proliferation, apoptosis, and cell motility [6]. Their altered expression and function in melanoma cells suggest a significant contribution to oncogenic transformation and metastatic potential.

TRPM1 (melastatin), originally identified as a melanocyte differentiation marker, is notably downregulated in advanced melanomas compared to benign nevi and normal melanocytes. Its expression is associated with a less aggressive phenotype and favorable prognosis, supporting its putative role as a tumor suppressor [27]. TRPM1-mediated calcium signaling appears to restrain cell proliferation and invasion, although its exact molecular functions in melanoma remain under investigation. Importantly, the TRPM1 gene locus also encodes microRNAs (e.g., miR-211), which are themselves implicated in melanoma progression and regulation of oncogenic pathways. It has been determined that both miR-211 and the TRPM1 channel protein are under the regulatory influence of the microphthalmia-associated transcription factor (MITF) at a transcriptional level [43]. Furthermore, it has been established that miR-211 directly targets potassium calcium-activated channel subfamily M alpha 1 (KCNMA1), a gene which is commonly associated with enhanced cell proliferation and increased migratory and invasive capabilities in various cancer types. The level of miR-211 was found to be reduced in almost all melanomas in comparison to melanocytes. This finding serves to confirm the tumor suppressor function of miR-211 [44,45]. The TRPM1 gene influences melanoma behavior through the generation of the following two distinct transcripts: the TRPM1 channel protein, encoded by its exons, and microRNA-211, located within its sixth intron [44]. This finding reflects the dual functional role of the gene, in that the loss of the TRPM1 protein serves as a marker of melanoma aggressiveness, while the expression of microRNA-211 has been associated with tumor-suppressing activity [24].

TRPM7 is frequently overexpressed in melanoma cells and is linked to enhanced proliferation, migration, and epithelial–mesenchymal transition (EMT). TRPM7 possesses both ion channel and serine/threonine kinase domains, enabling it to regulate diverse signaling cascades, including those involving magnesium and calcium flux, cytoskeletal remodeling, and cell adhesion [27,46]. Experimental silencing of TRPM7 has been shown to impair melanoma cell motility and reduce metastatic potential, highlighting its functional relevance [29,47,48,49].

TRPM8, a cold receptor, is of particular significance in the context of both melanocyte physiology and the development of melanoma. In a study by Yamamura et al. (2008) [33], it was demonstrated that TRPM8 is expressed in human melanoma cells (G-361 cell line), and its activation leads to a sustained influx of calcium ions. Kijpornyongpan et al. (2014) [50] discovered that the activation of the TRPM8 channel by its agonist, menthol, resulted in a cytotoxic effect in vitro on A375 human malignant melanoma cells. Higher doses of menthol induced distinct morphological changes in melanoma cells, including rounding, shrinkage, and even disruption of the cell membrane. These findings suggest that TRPM8 plays a key role in regulating the viability of A375 melanoma cells. The activation of TRPM8 has been demonstrated to disrupt intracellular calcium homeostasis, resulting in diminished cell proliferation. In the absence of corrective measures, this process can ultimately culminate in cell death [33,50,51]. Previous studies on different types of cancer, including lung, breast, and prostate carcinoma, have also shown that TRPM8 is expressed at higher levels in tumor tissues than in normal tissues. These findings support the involvement of TRPM8 overexpression in the development of cancer [52].

TRPV1 and TRPV4, members of the vanilloid subfamily, have been implicated in melanoma biology, though their roles remain somewhat controversial [39]. TRPV1 is upregulated in certain melanoma lines and may promote inflammation-associated carcinogenesis by modulating the tumor microenvironment and oxidative stress responses. Conversely, some studies suggest a potential anti-tumorigenic function through TRPV1 mediated apoptosis and the inhibition of proliferation. Yang et al. (2018) [53] demonstrated in their studies on melanoma cell lines (A2058 and A375) that increased TRPV1 expression significantly impairs the ability of cells to proliferate and clonogenicity. A similar effect was observed after the use of capsaicin, a potent TRPV1 agonist. This caused an increase in the expression of this channel in cancer cells, leading to the inhibition of their proliferation. Importantly, the use of the specific TRPV1 antagonist capsazepine reversed these effects, confirming that the action of capsaicin was mainly mediated by TRPV1. In subsequent stages of the study, it was demonstrated that TRPV1 activation resulted in a marked increase in p53 protein expression and its acetylation, which enhanced its transcriptional activity and led to the induction of apoptosis. In consideration of the findings, it can be posited that TRPV1 functions as a tumor suppressor in melanoma, thereby constraining proliferation and inducing the programmed cell death of tumor cells [36,53].

Similarly, TRPV4 activation has been reported to have influence on melanoma cell proliferation and survival. In the research conducted by Olivan-Viguera et al. [41] on A375 melanoma cells, the TRPV4 channel was found to be functionally expressed in these cells. The activation of TRPV4 using the selective agonist GSK1016790A has been shown to induce an immediate influx of Ca^2+^ ions and, often, co-activation of the KCa3.1 potassium channel. Consequently, there is a rapid dissolution of cell structure, accompanied by nuclear condensation, cell detachment from the substrate, and the initiation of apoptosis. The proliferation and survival of A375 cells are strongly inhibited [41]. Research by Li et al. (2022) [54] also revealed that TRPV4 is functionally overexpressed and activated by selective agonists (GSK1016790A and 4α PD) in A375 melanoma cells. This process has been shown to result in a substantial influx of calcium ions, which in turn leads to vesicular exocytosis. This triggers a series of morphological changes within the cells, in addition to mitochondrial dysfunction, ultimately resulting in ferroptosis (a form of programmed cell death that is dependent on iron) [54].

Emerging evidence suggests that other TRP channels, including TRPC1 and TRPA1, also contribute to melanoma pathophysiology. TRPC1 has been associated with calcium-dependent proliferation and resistance to apoptosis, while TRPA1 is activated by oxidative stress and electrophilic compounds, possibly influencing inflammatory pathways and cell survival [39,55]. However, the specific roles of these channels in melanoma are not yet fully elucidated and warrant further exploration.

Another TRP channel that plays a distinctive role in melanoma progression is TRPM2. Among these, TRPM2 also warrants attention due to its emerging role in melanoma biology [56,57]. TRPM2 is a calcium-permeable, oxidative stress-sensitive channel that is activated by ADP-ribose and reactive oxygen species (ROS) [58,59]. Its expression has been reported in various cancer types, including melanoma, where it contributes to tumor cell survival under oxidative and metabolic stress. Unlike other TRP channels that directly stimulate proliferation or migration, TRPM2 appears to support tumor cell viability by regulating mitochondrial function, redox homeostasis, and DNA damage responses. The inhibition or genetic silencing of TRPM2 has been shown to sensitize melanoma cells to oxidative stress and reduce their capacity to withstand cytotoxic insults, thereby enhancing the efficacy of chemotherapeutic agents [60]. This suggests that TRPM2 may serve as a metabolic and stress-response modulator in melanoma, and its blockade could represent a viable strategy to exploit the tumor’s vulnerability to redox imbalance. However, since TRPM2 also performs cytoprotective roles in non-malignant cells, achieving selective inhibition remains a key challenge.

Although many TRP channel family members are expressed in the skin, not all have been directly linked to melanoma progression. In recent years, other members of the TRP channel family have emerged as potential contributors to progression and skin tumor biology. It is evident that TRPC channels play a pivotal role in the regulation of calcium homeostasis. In particular, abnormal activation of TRPC5 has been identified as a potential driver of calcium imbalance within cancer cells. This disruption has been shown to activate signaling pathways associated with melanoma progression, especially in the context of developing resistance to chemotherapy. Additionally, TRPC5 has been implicated in the modulation of the tumor microenvironment, vascular remodeling, wound repair processes, and angiogenesis [61].

TRPC4, another canonical TRP channel, has also been associated with skin tumorigenesis. A recent immunohistochemical study by Kurz et al. [62] demonstrated TRPC4 protein expression in malignant melanoma, as well as in basal cell carcinoma and squamous cell carcinoma, suggesting broader relevance beyond melanocytic tumors. While the precise role of TRPC4 in melanoma remains to be fully elucidated, its consistent presence across multiple skin cancer types indicates involvement in calcium-mediated processes such as cell migration and invasion [62,63].

TRPV3, a calcium-permeable, nonselective cation channel, is abundantly expressed in skin keratinocytes. Although its role in cancer remains underexplored, increasing evidence suggests its participation in tumor development and progression [64]. Altered TRPV3 expression levels have been reported in several malignancies, including non-small cell lung cancer, melanoma, squamous cell carcinoma, and breast cancer. This upregulation has been associated with enhanced cancer cell proliferation, survival, and invasiveness [6,65]. These observations highlight TRPC4, TRPC5, and TRPV3 as emerging TRP channels of interest that merit further mechanistic exploration in melanoma biology.

In summary, distinct TRP channel isoforms exert either tumor-promoting or tumor-suppressive effects in melanoma, depending on their expression pattern and cellular context. The involvement of TRP channels in melanoma progression is illustrated in Figure 1. Understanding their specific contributions to signaling networks may reveal novel diagnostic biomarkers and therapeutic targets.

Recent studies have primarily focused on altered expression and functional changes of TRP channels in melanoma; however, knowledge about the prevalence of genetic alterations within these channels in clinical samples remains limited. In order to evaluate the mutational landscape of TRP channels in melanoma, the Skin Cutaneous Melanoma (TCGA, PanCancer Atlas) dataset was analyzed. This dataset comprised 442 patients and was investigated using cBioPortal (version 6.3.3; accessed 4 August 2025). Genetic alterations were identified in multiple TRP family members, as outlined in Table 1. These findings suggest that, although mutations in TRP channel genes are not prevalent, they are detectable in a significant proportion of melanoma cases [66].

The subsequent stage of the analysis entailed the examination of the correlation between these genetic changes and patient survival. The Kaplan–Meier survival analysis demonstrated that there were no statistically significant differences in overall survival between patients with mutations in any of the analyzed TRP channel genes and patients without mutations (*p*-value = 0.415) [66].

## 5. TRP Channels as Potential Therapeutic Targets

Transient receptor potential (TRP) channels have emerged as promising therapeutic targets in melanoma due to their involvement in various tumorigenic processes, including cell proliferation, apoptosis resistance, calcium homeostasis, migration, epithelial–mesenchymal transition (EMT), and response to oxidative and inflammatory stimuli [9,67]. Their localization at the plasma membrane, functional plasticity, and differential expression patterns between benign nevi and malignant melanoma enhance their relevance in the search for new therapeutic strategies. Importantly, the accessibility of TRP channels to pharmacological modulation, along with their ability to regulate intracellular signaling in response to external physical and chemical cues, makes them compelling candidates for anti-melanoma drug development.

Among the oncogenic TRP isoforms, TRPM7, TRPV1, TRPV4, and TRPM8 have received considerable attention due to their frequent overexpression in melanoma cells and their functional contributions to the malignant phenotype [20,24,68]. In vitro studies have demonstrated that the inhibition of these channels can reduce proliferation, suppress cell migration, promote apoptosis, and reverse resistance to conventional therapies. TRPM7, in particular, plays a central role in regulating intracellular Mg^2+^ and Ca^2+^ levels, cytoskeletal reorganization, and EMT-related signaling [69]. The pharmacological inhibition of TRPM7 using compounds such as waixenicin A or NS8593 has been shown to exert anti-proliferative and anti-metastatic effects not only in melanoma but also in other malignancies, underlining its potential as a broad-spectrum therapeutic target [70]. In comparison, TRPV1, the capsaicin sensitive ion channel, has demonstrated dual and context-dependent roles. On the one hand, its antagonism with agents like capsazepine can suppress inflammation-associated tumorigenesis and oxidative stress within the microenvironment of the tumor. On the other hand, activation of TRPV1 has been linked to pro-apoptotic signaling, indicating that its modulation must be carefully tailored to the tumor context [53,71,72]. Similarly, the blockade of TRPM8, a cold-sensitive calcium channel, has been reported to inhibit melanoma cell migration and enhance sensitivity to chemotherapy, likely through modulation of calcium-dependent survival pathways [73].

Together, these findings underscore the complex yet targetable roles of oncogenic and stress-adaptive TRP isoforms in melanoma, including TRPM2. Beyond its role in tumor adaptation, TRPM2 presents as an actionable target due to its selective activation under oxidative conditions [56,57]. Inhibiting TRPM2 has been shown to sensitize melanoma cells to redox imbalance and enhance responses to chemotherapeutic agents, positioning it as a potential metabolic checkpoint in melanoma therapy [74]. In contrast to the tumor-promoting functions of the above channels, TRPM1 is widely regarded as a tumor suppressor. Its expression is typically high in normal melanocytes and benign nevi but significantly reduced or absent in advanced melanomas, correlating with increased tumor aggressiveness and poor prognosis. Therapeutic strategies aimed at restoring or mimicking TRPM1 that function either directly or via the modulation of associated microRNAs—such as miR-211—are under investigation as potential interventions to curb melanoma progression [50,51,52,53]. Targeting upstream regulators of TRPM1 expression, such as the transcription factor MITF, may represent an indirect yet effective strategy to enhance its tumor-suppressive effects [23,39].

Nevertheless, the therapeutic targeting of TRP channels with dual or pleiotropic roles remains inherently complex. For instance, both TRPV1 and TRPM8 can exert either tumor-promoting or tumor-inhibiting effects, depending on the stage of the tumor, the composition of the tumor microenvironment, and the specific intracellular signaling milieu. Such context-dependent behavior calls for precise characterization of TRP expression profiles and functional states before therapeutic modulation is attempted [44,54,55].

The integration of TRP channel modulation into existing therapeutic frameworks offers additional avenues for improving treatment outcomes. In the era of immunotherapy and precision oncology, TRP channels could complement current regimens, such as BRAF/MEK inhibitors or immune checkpoint blockade. For example, the inhibition of TRP isoforms involved in chemoresistance, immune evasion, or neoangiogenesis may sensitize tumors to conventional therapies or potentiate immune responses. Furthermore, the ability of TRP channels to respond to physical stimuli such as temperature, mechanical stress, and voltage gradients introduces intriguing opportunities for combination therapies. Photothermal therapy, magnetothermal activation, or ultrasound-mediated drug delivery could be employed to spatially and temporally control TRP activity, allowing localized therapeutic effects with minimized systemic toxicity.

Despite their promise, several challenges must be addressed to fully harness TRP channels as therapeutic targets in melanoma. One major limitation is the current lack of highly selective TRP modulators. Many existing inhibitors and agonists display off-target effects or insufficient isoform specificity, limiting their clinical applicability. In addition, the heterogeneity of TRP channel expression among melanoma subtypes and across individual patients underscores the need for personalized, biomarker-driven approaches. Functional redundancy among TRP family members and compensatory signaling mechanisms may also diminish the efficacy of monotherapies targeting a single channel.

Moreover, much of the current evidence is derived from in vitro studies or non-melanoma models, and in vivo validation of TRP-targeted therapies in melanoma remains limited. Bridging this gap will require the development of robust animal models, the improved understanding of TRP channel regulation in physiological versus pathological states, and clinical studies assessing safety, pharmacokinetics, and the therapeutic window [75]. To advance the translational potential of TRP channel-based interventions, future research should prioritize the development of isoform-specific ligands, monoclonal antibodies, or RNA-based inhibitors tailored to the melanoma molecular landscape [23]. Parallel efforts should focus on mapping the downstream signaling pathways influenced by TRP activity, particularly those intersecting with immune regulation, metabolic reprogramming, and tumor–stromal interactions. The implementation of the high-resolution, spatiotemporal analysis of TRP channel expression in tumor biopsies and the use of TRP-targeting strategies in patient-derived xenograft models may further accelerate clinical translation.

In conclusion, TRP channels represent a versatile and underexplored class of therapeutic targets in melanoma. Their diverse functions in tumor progression, immunomodulation, and cellular homeostasis offer multiple opportunities for intervention, both as monotherapies and in combination with established treatment modalities. Continued efforts to understand and manipulate their roles will be critical for translating this emerging field into tangible clinical benefit for patients with melanoma.

## 6. Current Limitations and Future Directions in TRP Channel-Targeted Melanoma Therapy

Despite the growing body of evidence implicating TRP channels in melanoma biology, several important limitations hinder the immediate clinical translation of these findings. Most studies to date have been conducted in vitro, often using melanoma cell lines under controlled conditions that may not accurately reflect the complexity of the in vivo tumor microenvironment. As a result, the relevance of TRP channel modulation in actual patient tumors remains uncertain [11,67]. A major challenge is the lack of highly selective pharmacological modulators for individual TRP channel isoforms. Many currently available agonists or antagonists display limited specificity and off-target effects, complicating their potential use in therapeutic settings. Moreover, some TRP channels exhibit dual or context-dependent roles, acting as both tumor promoters and suppressors depending on tumor stage, intracellular signaling context, and microenvironmental factors. This functional plasticity demands a nuanced and highly personalized approach to therapeutic targeting.

Another significant gap is the paucity of in vivo studies and clinical trials investigating TRP channel-targeted therapies in melanoma. Although preclinical data are promising, there is a need for robust animal models and patient-derived xenografts to validate the therapeutic potential and safety of TRP modulators. Additionally, the development of isoform-specific antibodies, RNA-based inhibitors, or small-molecule modulators tailored to the molecular profile of individual tumors will be essential for successful clinical application. Future research should focus on mapping the spatiotemporal expression and activity of TRP channels in melanoma tissues, identifying biomarkers predictive of TRP dependency, and integrating TRP-targeting strategies with current treatment modalities such as immunotherapy, targeted kinase inhibitors, and redox-based therapies. The combination of TRP modulation with physical targeting methods, such as photothermal or ultrasound-mediated delivery, also warrants exploration to achieve precise, localized therapeutic effects [68].

To date, no clinical trials have been conducted to investigate the use of TRP channel modulators as a therapeutic agent for melanoma treatment. The available literature on these modulators has, until now, been limited to preclinical studies or early-phase clinical trials conducted on other tumors or diseases. To the best of our knowledge, no studies have yet incorporated melanoma patients. In summary, while TRP channels represent promising targets in melanoma therapy, addressing these limitations through advanced experimental models, improved drug design, and clinical investigation will be essential for translating this knowledge into effective patient care.

## 7. Conclusions

Transient receptor potential (TRP) channels have been demonstrated to play a key role in melanoma biology, influencing processes such as cation homeostasis, particularly calcium, proliferation, apoptosis, cell migration, and response to oxidative stress. Depending on the type of isoform, it can be demonstrated that these channels have the dual ability to function as both oncogenes and tumor suppressors. TRPM1 has been implicated in tumor suppression, while TRPM7, TRPV1, and TRPV4 have been observed to function as both melanoma suppressors and oncogenic drivers, modulating proliferation, apoptosis and metastasis. It has been established that TRPM1 is implicated in the suppression of tumors, while the other aforementioned TRP channels have been shown to exhibit dual functionality, acting as both melanoma suppressors and oncogenic drivers depending on the cellular context. The diverse and context-dependent roles of TRP channels underscore their potential as diagnostic biomarkers and therapeutic targets. However, the complexity of their function highlights the need for isoform-specific modulation and a personalized therapeutic approach. It is imperative that further research is conducted with a focus on the utilization of selective pharmacological tools, in vivo validation, and clinical studies. These endeavors are essential for the effective harnessing of the full therapeutic potential of TRP channels in the context of melanoma.

## Figures and Tables

**Figure 1 ijms-26-07829-f001:**
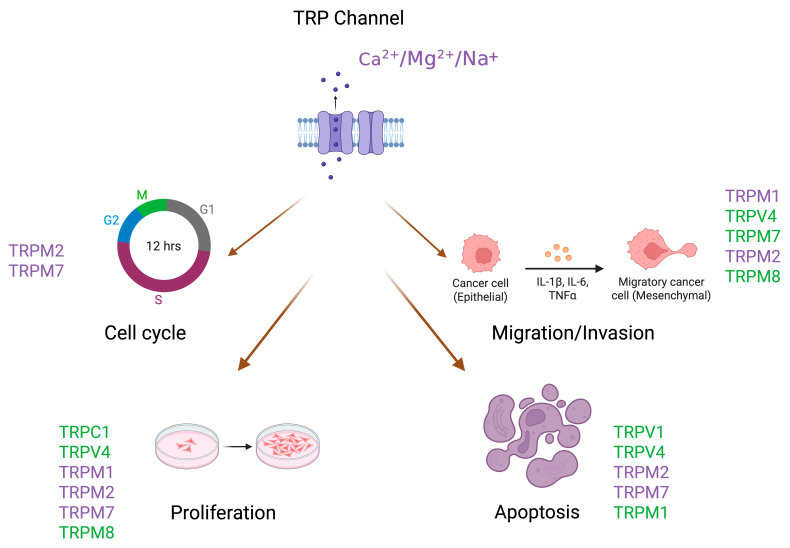
Role of selected TRP channels in regulating key cancer-related processes in melanoma. TRP channels marked in purple (TRPM1, TRPM2, TRPM7) exhibit tumor-promoting properties by supporting proliferation and migration and by inhibiting apoptosis. Channels marked in green (TRPM8, TRPV1, TRPV4) act as tumor suppressors by inhibiting proliferation and inducing apoptosis in melanoma cells. Arrows and symbols indicate the effects of individual channels on processes such as proliferation, migration/invasion, apoptosis, and cell cycle regulation. (Created in BioRender. Gagat, M. (2025) https://BioRender.com/u8k9zed accessed on 4 August 2025).

**Table 1 ijms-26-07829-t001:** Frequency of genetic alterations in TRP channel genes in patients with skin cutaneous melanoma (TCGA, PanCancer Atlas; n = 442). Data were retrieved from cBioPortal [66].

Gene Symbol	Percent Samples Altered (%)
*TRPM2*	14
*TRPM8*	9
*TRPM1*	8
*TRPM7*	8
*TRPV1*	5
*TRPV4*	5

## Data Availability

Not applicable.

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
