# Peer review of "TRP Channels in Skin Cancer: Focus on Malignant Melanoma"

_ijms, 2025, doi:10.3390/ijms26167829_

Round 1
Reviewer 1 Report
Comments and Suggestions for Authors
This is a well-written and very interesting paper on TRP channels in skin cancer. I only have a handful of suggestions to improve the work:
1) I wonder why channels like TRPV3 are missing. There are studies showing the presence in skin as well. Possibly, a couple of sentences on other (not thoroughly studied) TRP-channels could be added.
2) There have been recent studies von TRPC4 (Int J Mol Sci. 2023 Jan 5;24(2):1037) and TRPC5 (Biology (Basel). 2025 Jul 7;14(7):823) that also show expression patterns in melanoma. Could be added to literature and/or discussion.
3) As little is known about TRPM8, could you elaborate (maybe some sentences) on the info we have on its role in other skin cancers or linked diseases?
Author Response
Thank you very much for your insightful comments and valuable suggestions. We appreciate your careful review and have revised the manuscript accordingly to address the raised points. The revised text now more clearly defines the rationale for including TRPC4, TRPC5, and TRPV3, and better reflects the current state of evidence regarding their relevance to melanoma biology. We have also added a brief discussion of TRPM8 and its potential role in skin-related malignancies. All suggested clarifications have been incorporated into the manuscript to enhance its clarity and scientific rigor.
1) I wonder why channels like TRPV3 are missing. There are studies showing the presence in skin as well. Possibly, a couple of sentences on other (not thoroughly studied) TRP-channels could be added.
In response, we have extended the discussion to include TRPV3 as an example of a calcium-permeable, nonselective TRP channel abundantly expressed in skin keratinocytes. Although its role in cancer, including melanoma, remains underexplored, emerging data indicate altered expression levels in various malignancies, suggesting its involvement in tumor progression and invasiveness. We agree that highlighting such under-investigated channels enhances the comprehensiveness of the review.
2) There have been recent studies von TRPC4 (Int J Mol Sci. 2023 Jan 5;24(2):1037) and TRPC5 (Biology (Basel). 2025 Jul 7;14(7):823) that also show expression patterns in melanoma. Could be added to literature and/or discussion.
Both references have now been reviewed and cited in the revised manuscript. In response, we have added a few sentences referring to these studies, which support the expression of TRPC4 and TRPC5 in melanoma and strengthen the relevance of these channels in the context of calcium signaling and tumor biology. These additions help to better substantiate their inclusion as emerging TRP channels of interest.
3) As little is known about TRPM8, could you elaborate (maybe some sentences) on the info we have on its role in other skin cancers or linked diseases?
In response, we have added some details about TRPM8. TRPM8 has been implicated in skin physiology as well as in other cancer types. Moreover, overexpression of TRPM8 has been reported in several malignancies, including prostate, breast, and lung cancer, where it has been linked to processes such as proliferation, migration, and cell cycle regulation. These additions aim to provide a broader context for TRPM8 and to highlight its potential relevance to melanoma biology.

Reviewer 2 Report
Comments and Suggestions for Authors
In this manuscript, the author investigates the role of TRP channels in malignant melanoma. Although the topic itself is relevant, the novelty of the manuscript is limited. This is because similar review articles have already been published in recent years—for example, Reference 55 cited by the author, which was published in 2023. Moreover, upon reviewing the content, there appear to be few newly published original research articles (excluding reviews) on this topic within the past two years, which further limits the manuscript’s originality.
- For the TRP channel members discussed by the author, such as TRPM1 to TRPM8, what are their mutation rates in melanoma?
- Similarly, are there any statistical analyses showing whether mutations in these genes are associated with patient survival?
- Although the author mentions the challenges in clinical translation, I would still like to know: Are there any ongoing clinical trials involving TRP-targeting drugs? Alternatively, is there any research or data indicating whether the use of these drugs improves patient survival?
- Some of the information would be better presented in table format to enhance readability for the reader.
Author Response
We sincerely thank the Reviewer for their valuable and constructive comments. We fully agree with the points raised, which we consider important for improving the quality and clarity of our manuscript. We have carefully reviewed all suggestions and incorporated the appropriate changes accordingly. Additionally, we have updated the manuscript with relevant recent literature to enhance its novelty and scientific rigor.
1) For the TRP channel members discussed by the author, such as TRPM1 to TRPM8, what are their mutation rates in melanoma?
To address it, we analysed the Skin Cutaneous Melanoma (TCGA, PanCancer Atlas) dataset comprising 442 patients using cBioPortal. Our analysis revealed genetic alterations in several TRP family members, with mutation frequencies as follows: TRPM2 (14%), TRPM8 (9%), TRPM1 (8%), TRPM7 (8%), TRPV1 (5%), and TRPV4 (5%). Although mutations in these genes are not highly prevalent, they occur in a significant subset of melanoma cases. This data has been added as Table 1 in the revised manuscript.
2) Similarly, are there any statistical analyses showing whether mutations in these genes are associated with patient survival?
We thank the Reviewer for raising an important point about the clinical association of these mutations. Kaplan–Meier survival analysis performed on the TCGA dataset showed no statistically significant difference in overall survival between patients with mutations in any of the analysed TRP channel genes and those without such mutations (p = 0.415). This finding has been incorporated into the manuscript.
3) Although the author mentions the challenges in clinical translation, I would still like to know: Are there any ongoing clinical trials involving TRP-targeting drugs? Alternatively, is there any research or data indicating whether the use of these drugs improves patient survival?
We fully agree with the importance of clinical trials. Currently, no clinical trials specifically targeting TRP channels for melanoma therapy have been registered. Available literature on TRP modulators is limited to preclinical studies or early-phase clinical trials conducted in other tumor types or diseases. To our knowledge, there is no clinical evidence yet indicating improved patient survival in melanoma patients treated with TRP-targeting drugs. This clarification has been added to the manuscript.
4) Some of the information would be better presented in table format to enhance readability for the reader.
We agree with the Reviewer’s suggestion and have therefore organized key data on TRP channel gene mutation frequencies in melanoma into a new table (Table 1) within the manuscript toimprove clarity and reader accessibility.

Round 2
Reviewer 2 Report
Comments and Suggestions for Authors
I have no other suggestions.